# Disproportionate Vitamin A Deficiency in Women of Specific Ethnicities Linked to Differences in Allele Frequencies of Vitamin A-Related Polymorphisms

**DOI:** 10.3390/nu13061743

**Published:** 2021-05-21

**Authors:** Masako Suzuki, Tao Wang, Diana Garretto, Carmen R. Isasi, Wellington V. Cardoso, John M. Greally, Loredana Quadro

**Affiliations:** 1Department of Genetics, Albert Einstein College of Medicine, Bronx, NY 10461, USA; john.greally@einsteinmed.org; 2Department of Epidemiology and Population Health, Albert Einstein College of Medicine, Bronx, NY 10461, USA; tao.wang@einsteinmed.org (T.W.); carmen.isasi@einsteinmed.org (C.R.I.); 3Department of Obstetrics and Gynecology and Women’s Health, Stony Brook University Medical Center, Stony Brook, NY 11794, USA; dgarretto33@gmail.com; 4Columbia Center for Human Development, Department of Medicine, Columbia University Irving Medical Center, New York, NY 10032, USA; wvc2104@cumc.columbia.edu; 5Department of Food Science and Rutgers Center for Lipid Research, and New Jersey Institute for Food, Nutrition, and Health, Rutgers University, New Brunswick, NJ 08901, USA; lquadro@sebs.rutgers.edu

**Keywords:** vitamin A deficiency, pregnant women, Hispanics/Latinos, NHANES, CDC

## Abstract

**Background:** While the current national prevalence rate of vitamin A deficiency (VAD) is estimated to be less than 1%, it is suggested that it varies between different ethnic groups and races within the U.S. We assessed the prevalence of VAD in pregnant women of different ethnic groups and tested these prevalence rates for associations with the vitamin A-related single nucleotide polymorphism (SNP) allele frequencies in each ethnic group. **Methods:** We analyzed two independent datasets of serum retinol levels with self-reported ethnicities and the differences of allele frequencies of the SNPs associated with vitamin A metabolism between groups in publicly available datasets. **Results:** Non-Hispanic Black and Hispanic pregnant women showed high VAD prevalence in both datasets. Interestingly, the VAD prevalence for Hispanic pregnant women significantly differed between datasets (*p* = 1.973 × 10^−10^, 95%CI 0.04–0.22). Alleles known to confer the risk of low serum retinol (rs10882272 C and rs738409 G) showed higher frequencies in the race/ethnicity groups with more VAD. Moreover, minor allele frequencies of a set of 39 previously reported SNPs associated with vitamin A metabolism were significantly different between the populations of different ancestries than those of randomly selected SNPs (*p* = 0.030). **Conclusions:** Our analysis confirmed that VAD prevalence varies between different ethnic groups/races and may be causally associated with genetic variants conferring risk for low retinol levels. Assessing genetic variant information prior to performing an effective nutrient supplementation program will help us plan more effective food-based interventions.

## 1. Introduction

The essential micronutrient vitamin A plays critical roles in vision [1,2], the immune system [3,4], cell growth and differentiation [5,6,7,8], as well as in the development of multiple organs, including the lungs, heart, eyes, and kidneys (reviewed in [9,10]). Since vitamin A is an essential micronutrient, limited access to vitamin A-rich food or supplements severely affects the tissue and blood levels of vitamin A in humans [11]. While the current prevalence rate of vitamin A deficiency (VAD)—defined as serum retinol levels lower than 1.05 µmol/L—is estimated to be less than 1% in the developed countries [12,13], it has been suggested that overall rates might obscure differences in ethnic groups and races, even in wealthy, developed countries [14,15].

Recently, we reported a high VAD prevalence among pregnant women in the Bronx, NY, USA [16], where the ethnic diversity and the poverty rate is much higher than in the rest of the nation [17,18]. Although the original study addressed the effects of bariatric surgery on serum vitamin A levels during pregnancy, the most surprising result was that more than 60% of the pregnant women who did not undergo the bariatric surgery (the control group) had serum retinol levels during the third-trimester lower than 1.05 µmol/L [16], meeting the criteria for vitamin A deficiency. This proportion of vitamin A deficient women in the Bronx was much higher than that of non-White women (Hispanic/Latino, non-Hispanic Black and other race/ethnicity than non-Hispanic White) of the same age group in the U.S. recently reported by Hanson et al. [19]. In the study, the authors recruited Hispanic/Latino, non-Hispanic White, non-Hispanic Black, and other race/ethnicity (including multiracial) pregnant women to their study from the Labor and Delivery unit in a Midwestern United States Academic Medical Center, which is located in suburban Chicago, IL, USA, where Latin Americans with Mexican ancestry are the majority group of Hispanics/Latinos. As Latin Americans with Afro-Caribbean ancestry (39.4% Dominicans and 36.4% Puerto Ricans, U.S. Census data) are the major Hispanic/Latino populations in the Bronx [18], these differences prompted us to reanalyze the disproportionality of VAD status of pregnant women from different ethnic groups and races, and to test the associations of genetic variants with the VAD variability between the ethnic groups. Establishing an association with genetic variants in vitamin A processing genes could be of utmost importance for understanding the population risk of developing VAD in the United States and for planning interventions aimed at eradicating VAD.

## 2. Materials and Methods

### 2.1. Data Source and Study Population for Serum Retinol Levels

We used two independent datasets of serum retinol levels, the data from the Bronx study [16] and the National Health and Nutrition Examination Survey (NHANES). The Bronx dataset had been collected for a study addressing the effects of bariatric surgery on serum vitamin A levels during pregnancy [16]. Women planning to breastfeed with singleton pregnancies with and without a history of bariatric surgery (either Roux-en-Y or gastric sleeve) were recruited [16]. Demographic data included race/ethnicity, education, prepregnancy and at delivery body mass index, gestational weight gain and parity. Pregnancy outcomes included gestational age at delivery, mode of delivery, and neonatal weight [16]. For the NHANES dataset, we used data from female participants in three NHANES cycles (2001–2002, 2003–2004, and 2005–2006), which followed a stratified, multistage, clustered probability sampling design. NHANES obtained written, informed consent for all participants, and the survey protocol was approved by the Research Ethics Review Board of the National Center for Health Statistics (NCHS). Detailed information on NHANES and the data are publicly available on the NHANES website (https://www.cdc.gov/nchs/nhanes, 30 March 2020). A total of 24,715 individuals provided information to NHANES in the three cycles. In this study, we focused on female participants (pregnant and nonpregnant) aged 17 to 42 years (4662 individuals). After excluding those who did not provide serum retinol levels, family poverty income ratios, self-reported ethnicity and information on important variables considered for the multivariable regression models (580 individuals), a total of 4082 study participants were included in the analysis.

### 2.2. Genetic Variations of the Vitamin A-Related Gene Polymorphisms

To assess potential genetic contributions to vitamin A status, we assessed the allele frequency of the vitamin A-related gene polymorphisms [20,21,22] between ethnic groups from publicly available population allele frequency datasets, such as the Allele Frequency Aggregator (ALFA) [23], the Population Architecture using Genomics and Epidemiology (PAGE [24], BioProject Accession: PRJNA168052), and the 1000 Genomes [25] projects. The ALFA pipeline consists of the allele frequency for variants in dbGaP that includes genomic data with subjects from 12 diverse populations worldwide [23]. The PAGE consortium dataset includes genetic variation data representing seven ethnic groups, i.e., African Americans, Asian Americans, American Indians, European Americans, Hispanic Americans (Latin Americans with Mexican ancestry and with Afro-Caribbean ancestry), and Native Hawaiians, from the United States-based cohorts [24]. Importantly, the Hispanic/Latino cohorts in PAGE contain subjects from the Hispanic Community Health Study/Study of Latinos (HCHS/SOL), with one of the HCHS/SOL research centers located in the Bronx, NY [24,26]. The associations of genetic variants with gene expression levels, defining expression quantitative trait loci (eQTL), were obtained from the Genotype-Tissue Expression project, GTEx [27].

### 2.3. Combinations of Low-Serum Retinol Risk Alleles

We obtained the genetic variant combinations of two well-known serum retinol level-associated polymorphisms, rs738409 and rs10882272, from the Trans-Omics for Precision Medicine (TOPMed) database (https://www.nhlbi.nih.gov/science/trans-omics-precision-medicine-topmed-program, 9 March 2021). We only included (1) control individuals from case-control studies and (2) nondiabetic and nonhypertensive individuals from family studies. A total of 9081 samples grouped by the reported ethnicities and race were used in the current analysis. The number of subjects used, the reported ethnicities and race, and exclusion criteria are shown in Appendix A.

### 2.4. Statistics Analysis

Appropriate sample weights were applied to account for complex survey designs and the unequal probability of selection, noncoverage, and nonresponse bias for calculating serum retinol variation between ethnicities. We classified the participants whose serum retinol levels were less than 1.05 µmol/L as vitamin A deficient. To assess the covariates between vitamin A deficient and sufficient participants, we used Student’s t-test for continuous variables, and Fisher’s exact test for categorical variables. The prevalence of deficiency in participants was estimated with proportions and 95% confidence interval (CI). To assess the significance level of the deviations of allele frequency of previously reported vitamin A-related gene polymorphisms between the ethnic groups compared to background noise, we used a permutation test with 1000 iterations. An alpha of 0.05 was used as the cutoff for significance. All statistical analyses were performed using R, The R Project for Statistical Computing (https://www.r-project.org/, 19 May 2021), version 4.0.2 [28].

## 3. Results

### 3.1. Differences in the Proportion of Vitamin A Deficiency in Pregnant Women between Different Ethnic Groups

#### 3.1.1. The Bronx Study

We used the third-trimester serum retinol levels that were available for 67 women out of the 96 participants in the Bronx study [16]. We found no significant association between missing data status and other covariates. While maternal serum levels of beta-carotene, the most abundant dietary vitamin A precursor [11], and cord blood serum retinol were significantly associated with maternal VAD status (*p* = 0.041 and *p* = 0.007, respectively), other known covariates, including the bariatric surgery status, did not show significant associations with maternal VAD status (Table 1). We then reanalyzed the third trimester serum retinol levels by ethnic group, specifically using the self-reported ethnicities (non-Hispanic Black, Hispanic/Latino, or other race) [16]. Our results showed that the proportion of VAD in Hispanic/Latino women was 65.9% (29 out of 44 Hispanic participants), in non-Hispanic Blacks was 53.3% (8 of 15 African American participants), and for other ethnicities was 37.5% (3 in 8 participants) (Table 1). Among the Hispanic/Latino participants (*n* = 44), vitamin A deficient women tended to be younger than the vitamin A sufficient women (*p* = 0.088), but education levels, prepregnancy body mass index (BMI), and gestational weight gain (GWG) were not associated with the VAD status (*p* = 0.876, *p* = 0.195, and *p* = 0.935, respectively). The degree of education and poverty level are generally negatively correlated in the Bronx, NY (the poverty rate in those with educational attainment of less than High School is 36.6%, High School is 23.4%, and College is 19.5%, 2019 1-year estimates, U.S. Census) [18]. Therefore, we inferred that the poverty levels in the Bronx study were likely similar between vitamin A deficient and sufficient women, and that the poverty level might not have been directly associated with VAD status between ethnicities, in this cohort.

#### 3.1.2. NHANES

In the NHANES dataset, we used serum retinol values, poverty income ratio (PIR), pregnancy status and ethnicity/race from 4082 women aged 17–42 years old. Among those women, the proportions of VAD in each ethnicity ranged from 3.6% (non-Hispanic White) to 13.1% (non-Hispanic Black). The VAD proportions were significantly increased among pregnant women in all ethnic groups (Table 2). Moreover, as expected from the previous report [15], we observed a further increase in pregnant women with PIR < 1.85 (Table 2). According to the U. S. Census Bureau the PIR of 1.0 defines the minimum income needed to avoid poverty and thus it is used as measure of poverty threshold [18]. Blumberg et al. [29] reported that individuals in the poverty income ratio (PIR) > 1.85 subgroup had a lower prevalence of inadequacy for most nutrients, compared to the subgroups PIR ≤ 1.85 [29]. Interestingly, the proportions of VAD in non-Hispanic Black pregnant women with PIR < 1.85 were not significantly different between the NHANES and the Bronx datasets (*p* = 0.26, 95%CI 0.14–1.74), however, the proportions of VAD in Mexican American (Latin Americans with Mexican ancestry) pregnant women with PIR < 1.85 from the NHANES dataset were significantly lower than those of Hispanic pregnant women in the Bronx dataset (*p* = 2.232 × 10^−10^, 95%CI 0.04–0.22).

### 3.2. Genetic Variations and Differences of Vitamin A Deficiency Proportions

#### 3.2.1. Differences of Allele Frequencies of Serum Retinol Levels Associated SNPs between Different Ethnic Groups

In the GWAS Catalog [30], two single nucleotide polymorphisms, (SNPs) (rs10882272 T/C and rs1667255 C/A), identified from a genome-wide association study of 5006 “Caucasian” males, are listed as associated with serum retinol levels [22]. rs10882272 is located in the 3′ UTR of the free fatty acid receptor 4 (*FFAR4*) gene and downstream of the *RBP4* gene and rs1667255 is located between the downstream of the Transthyretin (*TTR*) gene and the downstream of the beta-1,4-galactosyltransferase 6 (*B4GALT6*) gene. To assess if these two genetic polymorphisms were associated with the VAD prevalence differences between different racial/ethnic groups, we first compared the low serum retinol allele frequencies of rs10882272 and rs1667255 between different ethnic groups in ALFA and PAGE. While we did not observe significant differences in major allele frequencies of rs1667255 between Hispanic/Latino groups (Appendix A), the allele frequencies of rs10882272 showed significant variation between different ethnic groups, as we predicted. The frequency of the allele associated with low serum retinol (rs10882272: frequency for C allele) was much higher in African (0.620, 62%) and African American (0.617) compared to European (0.383) and Asian (0.106) individuals in the ALFA dataset (Figure 1a). Similarly, the PAGE dataset results showed that the risk allele frequencies were higher in Latin Americans with Afro-Caribbean ancestry (Puerto Ricans (0.455), Dominicans (0.502) and Cubans (0.410)) compared to Mexicans (0.260), Central Americans (0.288), South Americans (0.278) or Native Americans (0.357) (Figure 1b). Furthermore, we tested whether the rs10882272 variant was an expression quantitative trait locus (eQTL) for its nearby genes, using a publicly available database (the Genotype-Tissue Expression project, GTEx) [27]. In the GTEx data, we detected the associations between the rs10882272 variant and the expression levels of RBP4 in the liver, where the gene is highly expressed [31], with the presence of the allele associated with low serum retinol levels also associated with increased expression of *RBP4* (normalized effect size: 0.137, *p* = 0.00012, and m-value 0.987). For *FFAR4*, we detected the association in the lungs (normalized effect size: 0.126, *p* = 8.5 × 10^−6^, and m-value 1.00), but not in the pituitary (normalized effect size: 0.0334, *p* = 0.5, and m-value 0.809). The pituitary showed the highest expression of *FFAR4* in the GTEx data.

Another polymorphism, rs738409, which is located in the coding sequence of the patatin-like phospholipase domain containing 3 (*PNPLA3*) gene, has also been reported the be associated with serum retinol levels [21]. *PNPLA3* encodes a gene involved in the mobilization of retinyl esters stored in stellate cells [21,32]. The rs738409 polymorphism is a missense variant, with the C to G nucleotide substitution changing the amino acid I[ATC] to M[ATG]. The frequency of the mutant allele varies between ethnic groups (African American (0.144), Asian (0.44), Cuban (0.28), Dominican (0.26), Mexican (0.50) and Puerto Rican (0.34); PAGE dataset, Appendix A).

#### 3.2.2. Proportions of Low-Serum Retinol Higher-Risk Populations Vary between Race/Ethnic Groups

Next, we assessed the combinations of the polymorphisms in each race/ethnic group using TOPMed datasets (Appendix A). We combined nine case-control and family studies to the analysis, based on the reported race/ethnicity. We excluded cases and individuals with diabetes or hypertension to avoid unknown sampling bias. In total, we included 9081 individuals with 1808 Asian, 1610 non-Hispanic White (European), 1487 non-Hispanic Black (African American), 3656 Mexican and 520 Afro-Caribbean (Latin American, African Caribbean) participants. We plotted the proportions of each polymorphism genotype by race/ethnic group (Figure 2a,b). We observed higher proportions of the homozygous rs10882272 risk alleles (C/C) in non-Hispanic Blacks and Afro-Caribbeans (Figure 2a), and in contrast, Asians and Mexicans had higher proportions of homozygous rs738409 risk alleles (G/G) (Figure 2b). In Figure 2c, we show the proportion of each combination of these two SNPs. The population with low-risk homozygous genotypes (rs738409 C/C and rs10882272 T/T) are high among Asians and non-Hispanic White. While the numbers of individuals homozygous for both risk alleles are small, the highest was Mexican (1.53%) followed by Afro-Caribbeans (0.96%). When we considered at least one of the high-risk allele homozygous as higher risk populations, 37.93% of non-Hispanic Blacks and 40.96% of Afro-Caribbean’s were classified as higher risk populations (Figure 2d).

#### 3.2.3. Variations of Allele Frequencies of Polymorphisms Associated with Serum Retinol and Beta-Carotene Levels as Well as with the Beta-Carotene Bioactivities between Race/Ethnicity Groups

In addition to the three above-mentioned SNPs, several GWAS and candidate gene association studies have identified other polymorphisms associated with serum retinol and beta-carotene levels as well as with beta-carotene bioactivities [20]. We therefore also assessed the allele frequency deviations of the 39 SNPs associated with circulating vitamin A levels [20] between different ethnic groups in the 1000 Genomes Project [25]. The deviations of allele frequencies of those vitamin A-related SNPs between different ethnic groups are listed in the Appendix A. The average of the allele frequency standard deviation among ethnic groups was 0.122, significantly higher than randomly selected sets of 39 SNPs from the 1000 Genomes data (*p* = 0.030, permutation test with 1000 iterations, Figure 3).

## 4. Discussion

In this study, we have been able to demonstrate that the VAD proportions in pregnant females vary among different race/ethnicity groups even in a wealthy, developed country. The proportions of VAD women in non-Hispanic Black and Hispanic/Latino groups were significantly higher than those of non-Hispanic White women, even in pregnant women living in poverty. Interestingly, although the proportions of VAD in non-Hispanic Black pregnant women living in poverty were comparable between the datasets, the proportions of VAD in Hispanic/Latino females were significantly different. Of note, while the major origins of Hispanic/Latino populations in the Bronx are Latin Americans with Afro-Caribbean ancestry (39.4% Dominicans and 36.4% Puerto Ricans, U.S. Census data [18]), the Hispanic/Latino participants of the NHANES were Latin Americans with Mexican ancestry. Thus, these data strongly argue for an ancestry/ethnicity-specific impact on serum retinol levels and vitamin A status of pregnant women.

Serum retinol level differences among three Hispanic/Latino groups has been reported using the Hispanic Health and Nutrition Examination Survey (HHANES) conducted from 1982–1984 [33]. In contrast to our findings, the HHANES study found that Latin Americans with Mexican ancestry had a higher VAD prevalence rate than Latin Americans with Afro-Caribbean ancestry in both adults and children. However, this study was performed almost four decades ago when the participants’ nutrient status might have differed from the current one. Thus, the lower serum retinol levels reported in 1982–1984 in Latin Americans with Mexican ancestry might reflect the lower dietary intake of vitamin A at that time rather than genetic variations. Indeed, the CDC’s Second Nutrition Report in 2012 showed that serum retinol and retinyl ester concentrations increased in the general population in the United States between 1999–2000 and 2005–2006 (geometric mean, retinol: 52.8 µg/dL to 54.7 µg/dL and retinyl ester: 1.23 µg/dL to 2.11 µg/dL). Interestingly, serum retinyl palmitate levels, an indicator of newly ingested vitamin A, increased at a greater magnitude in Latin Americans with Mexican ancestry (geometric mean 0.759 ug/dL in 1999–2000 to 1.85 ug/dL in 2005–2006), although this latter group still showed the lowest retinyl ester concentrations compared to other ethnic groups [34]. An updated population-based measurement of serum vitamin A levels is clearly needed.

Since vitamin A is an essential micronutrient, the deficiency status is strongly associated with food insecurity. Thus, improving the nutritional status of pregnant women is the best way to prevent maternal micronutrient deficiency, however, our results suggested that underlying genetic variation is also associated with micronutrient deficiency status in these groups. Genetic contributions to the levels of circulating retinol have been reported in European populations. Specifically, a family study in France showed that the heritability estimate for serum retinol concentration (30.5%) was much larger than the variability accounted for by household, i.e., individuals living in the same house (14.2%) [35]. There are three well-identified low-serum retinol SNPs in the literature, rs1667255, rs10882272, and rs738409 [21,22]. The first two SNPs, rs1667255 C/A and rs10882272 T/C, were identified from a genome-wide association study of 5006 males, and the association of rs10882272 was replicated in independent samples, including 3792 females and 504 males [21]. While the authors were not able to test the association of these variants with VAD status, as the rate of VAD in the individuals studied was low, however, they reported that the estimated relative difference in mean retinol levels per copy of the rs10882272 C allele from the overall meta-analysis was decreased by 3.0% [21]. The rs1667255 variant is located between the downstream of Transthyretin (*TTR*) gene and the downstream of beta-1,4-galactosyltransferase 6 (*B4GALT6*) gene. Transthyretin forms a ternary complex with RBP4 and retinol that transport vitamin A in the bloodstream [36]. The rs10882272 variant is located in the 3′ UTR of the free fatty acid receptor 4 (*FFAR4*) gene and downstream of the *RBP4* gene. The *FFAR4* gene encodes a GPCR receptor (GPCR120) for free long-chain fatty acids, including omega-3 [37,38]. *FFAR4*/GPCR120 is expressed in various cell types, including the pituitary, lungs, macrophages, adipocytes, intestinal neuroendocrine cells and pancreatic cells. Thus, it participates in a number of physiological processes, including energy regulation, insulin sensitivity, immunological homeostasis and anti-inflammatory responses [39]. RBP4 is the sole specific carrier for retinol in the bloodstream [2,40]. Predominantly expressed in hepatocytes, RBP4 binds retinol to mobilize vitamin A from the liver, the primary body storage site of the vitamin, towards the peripheral tissues [2]. Interestingly, the proportion of lower serum retinol-risk allele frequencies (rs10882272 C) were higher in non-Hispanic Blacks and Afro-Caribbeans, which have higher VAD proportions. The third SNP rs738409, located in the *PNPLA3*, is a missense variant with the C to G nucleotide substitution changing the amino acid I[ATC] to M[ATG] [21]. *PNPLA3* encodes a gene involved in the mobilization of retinyl esters stored in stellate cells [21,32]. The *PNPLA3* I148M missense variant is a loss-of-function mutation, and individuals homozygous for *PNPLA3* I148M have lower circulating levels of RBP4 as well as lower serum retinol levels [21]. A significantly high correlation between the serum concentrations of retinol and those of RBP4, even in the context of acute phase response and protein malnutrition, has been reported [41]. Therefore, the PNPLA3 I148M homozygotes may have lower serum retinol levels compared to 148I allele carriers. In animal models, while retinol deficiency leads to an accumulation of RBP4 in liver, likely by inhibiting its secretion from this organ, hepatic RBP4 mRNA levels show no differences between vitamin A deficiency and sufficiency [42,43]. Of note, the *PNPLA3* I148M missense variant and the risk of nonalcoholic fatty liver disease (NAFLD) have been associated [44,45,46]. In contrast to the rs10882272 and rs1667255 variants which were identified from non-Hispanic White/European participants, a higher proportion of the rs1667255 *PNPLA3* I148M missense variant has been reported among Mexicans [46,47]. On the PAGE data, Mexican has the highest frequency of the missense variant (0.50) followed by Asian (0.44), Puerto Rican (0.34), Cuban (0.28), Dominican (0.26), and African American (0.144).

The proportions of lower serum retinol-risk alleles (rs10882272 C and rs738409 G) vary among different race/ethnic groups and the proportions tend to be higher in groups with a higher prevalence of VAD (non-Hispanic Blacks and Hispanics/Latinos). To assess the combinations of these two variants between race/ethnic groups, we used genotype data of control individuals from case-control studies and participants without type 2 diabetes or hypertension from family studies of TOPMed. The goal of the TOPMed program is to generate resources that can improve the understanding of heart, lung, blood and sleep disorders and advanced precision medicine. While we excluded participants with diseases, maternal VAD is associated with increased risks of lung and immune system problems for the offspring later in life [3,48,49]; our detailed analyses to assess genetic variations and these phenotypes are ongoing. As we expected, we replicated the highest allele frequency of rs10882272 C in non-Hispanic Blacks (0.61) and of rs738409 G in Mexicans (0.45). We calculated the combinations of these variants and found that the homozygosity for the risk alleles for both polymorphisms occurred infrequently in all race/ethnicity groups, with the highest rates in Mexicans (1.53%) followed by Afro-Caribbeans (0.96%). While the rs738409 G variant also showed a higher proportion in Asians (0.39), the population with no risk homozygous genotypes (rs738409 C/C and rs10882272 T/T) was high among Asians and non-Hispanic Whites. When we considered at least one of the high-risk allele homozygous as indicating higher risk populations, 40.96% of Afro-Caribbeans, 37.93% of non-Hispanic Blacks, 26.39% of Mexicans, 16.76% Asians and 16.21% of non-Hispanic whites are classified as higher risk. These findings suggest that non-Hispanic Blacks and Latin Americans with Afro-Caribbean ancestry have a higher risk of low-serum vitamin A for genetic reasons.

We acknowledge that there are several limitations to our study: the sample size of the Bronx cohort (*n* = 97) was limited, the poverty levels and clinical information on VAD-related outcome of the Bronx cohort are not available, the serum retinol data of NHANES were collected almost two decades ago, and the genotype information of all participants is not available in both cohorts. While the TOPMed provides us with genotypic and phenotypic information, none of studies measured the vitamin A status of the participants. Since the overall prevalence of VAD is considered to be rare in developed countries, vitamin A levels have not been measured routinely and populations levels of the vitamin are not up to date [50]. Further genome-wide association studies with demographic information, including food accessibility/intake in multiethnic cohorts, are needed to assess the influences of genetic variation and the different VAD statuses between different ethnic groups. Moreover, while the WHO does not recommend routine vitamin A supplementation to pregnant women to avoid excessive intake, they recommend vitamin A supplementation to pregnant women in a given geographical area if ≥20% of pregnant women have serum retinol levels <0.70 µmol/L [51]. Our reanalysis of the Bronx study showed that more than 40% of pregnant women had serum retinol <0.70 µmol/L, strongly suggesting that urgent action needs to be taken to reduce VAD, especially in unusually susceptible ethnic groups, to reduce the risk of adverse health conditions for the mother [52] and diseases for the offspring later in life [48,53]. Assessing genetic variant information prior to performing an effective nutrient supplementation could be a critical strategy to plan more effective food-based interventions to reduce VAD.

## 5. Conclusions

While VAD in developed countries is generally believed to be a rare condition, there is a substantial proportion of pregnant women of certain ethnic groups who suffer from VAD, even in wealthy, developed countries. Moreover, our results showed that genetic polymorphisms may be contributing to the VAD status differences between ethnic groups, at least in the pregnant women we studied. As aforementioned, the prevalence of VAD is strongly associated with the intake of provitamin A or retinyl esters from food. Hence, improving micronutrient intake is the best option to reduce VAD in the population. However, our results suggest that genetic variants might also be associated with VAD status. Therefore, we propose that the genetic variant information be taken into consideration to perform an effective nutrient supplementation program. Further understanding of this association will ultimately enable adequate food interventions based on the genetic information that could be crucial to improve maternal vitamin A status during pregnancy in these higher-risk groups.

## Figures and Tables

**Figure 1 nutrients-13-01743-f001:**
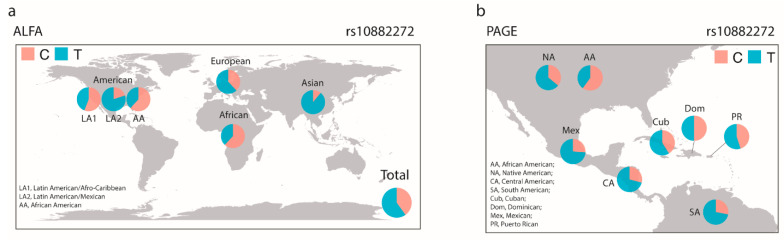
Variations of the allele frequencies of rs10882272. We plotted the allele frequency of each ethnic group (**a**) Allele Frequency Aggregator (ALFA) and (**b**) Population Architecture using Genomics and Epidemiology (PAGE). C, rs10882272 C allele; T, rs10882272 T allele.

**Figure 2 nutrients-13-01743-f002:**
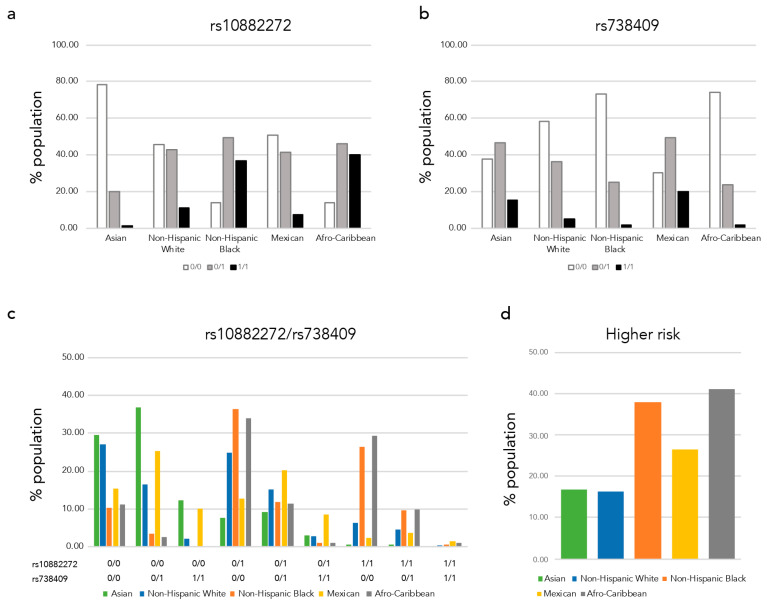
Low-serum retinol risk allele proportions of each race/ethnic group. We plotted the low-serum retinol risk allele proportion of each ethnic group (**a**) rs1088272, (**b**) rs738409, (**c**) rs1088272 and rs738409 variant combinations, and (**d**) proportion of higher risk carrier.

**Figure 3 nutrients-13-01743-f003:**
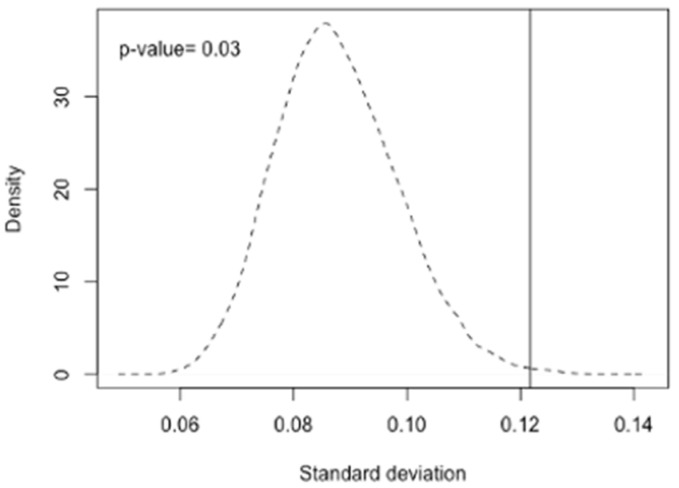
Permutation analysis of the deviation of MAF between different ethnic groups. We plotted the averaged distribution of standard deviation of MAF of randomly selected 39 SNPs from 6,025,429 SNPs 1000 times with a dashed line. The vertical line indicates the mean value of vitamin A related SNPs (0.121). The result suggests that deviation of the allele frequencies of the serum retinol related SNPs among different ethnicities is significantly larger than that of randomly selected SNPs (*p*-value = 0.03).

**Table 1 nutrients-13-01743-t001:** Demographic information of the participants by serum retinol levels of the Bronx study.

Participants (*n*)		VAD * (*n* = 40)	VAS † (*n* = 27)	*p* Trend
Ethnicity, *n* (%)	Hispanic/Latin American	29 (72.5)	15 (55.6)	0.311
	non-Hispanic Black	8 (20.0)	7 (25.9)	
	Other	3 (7.5)	5 (18.5)	
Age (y)		29.1 (5.4)	31.3 (6.1)	0.138
Multipara, *n* (%)	No	10 (25.0)	7 (26.9)	1.000
	Yes	30 (75.0)	19 (73.1)	
Bariatric surgery, *n*(%)	No	21 (52.5)	13 (48.1)	0.806
	Yes	19 (47.5)	14 (51.9)	
Education level, *n* (%)				0.458
	<High school	5 (13.2)	4 (16.7)	
	High School	22 (57.9)	10 (41.7)	
	College	11 (28.9)	10 (41.7)	
Body Mass Index (kg/m^2^)	Prepregnancy	31.8 (7.6)	32.1 (7.4)	0.878
	At delivery	36.0 (7.1)	36.9 (6.3)	0.619
Obese (prepregnancy), *n* (%)				0.952
	Normal Weight	5 (12.8)	3 (11.5)	
	Over Weight	9 (23.1)	8 (30.8)	
	Class I obesity	13 (33.3)	8 (30.8)	
	Class II obesity	12 (30.8)	7 (26.9)	
Gestational weight gain (GWG)		24.4 (20.1)	26.2 (12.3)	0.683
GWG recommended level, *n* (%)	Below	5 (13.2)	2 (7.7)	0.691
	Above	33 (86.8)	24 (92.3)	
Delivery mode (C-section), *n* (%)	No	24 (61.5)	12 (46.2)	0.309
	Yes	15 (38.5)	14 (53.8)	
Baby weight (g)		3053.8 (602.9)	3104.4 (601.1)	0.742
Large for genstational age, *n* (%)	No	32 (97.0)	22 (95.7)	1.000
	Yes	1 (3.0)	1 (4.3)	
Serum retinol (µmol/L)	Postpartum (mother)	0.5 (0.4)	0.5 (0.6)	0.886
	Cord blood	0.2 (0.1)	0.3 (0.1)	0.041
Serum beta carotene (ng/100 µL)	Third trimester (mother)	3.3 (2.0)	6.2 (5.9)	0.007
	Postpertum (mother)	2.6 (2.1)	3.1 (1.7)	0.413
	Cord blood	1.7 (0.6)	2.0 (0.6)	0.554

*p* values were obtained from t test or chi-square test, where appropriate, VAD *; vitamin A deficiency, VAS †; vitamin A sufficiency.

**Table 2 nutrients-13-01743-t002:** VAD proportions of 4082 women in NHANES datasets by self-reported ethnic groups.

		Non-Hispanic White			Non-Hispanic Black			Mexican American			Other Hispanic			Other		
		Total *	Pregnant	Pregnant with PIR <1.85	Total *	Pregnant	Pregnant with PIR <1.85	Total *	Pregnant	Pregnant with PIR <1.85	Total *	Pregnant	Pregnant with PIR <1.85	Total *	Pregnant	Pregnant with PIR <1.85
Total N (%)		1684 (41.3)	392 (45.5)	115 (26.9)	976 (23.9)	145 (16.8)	91 (21.3)	1071 (26.2)	239 (27.8)	181 (42.3)	168 (4.1)	36 (4.2)	22 (5.1)	183 (4.5)	49 (5.7)	19 (4.4)
Vitamin A status	Deficient	60 (3.6)	43 (11.0)	14 (12.2)	128 (13.1)	47 (32.4)	33 (36.3)	72 (6.7)	36 (15.1)	29 (16.0)	15 (8.9)	10 (27.8)	6 (27.3)	14 (7.7)	6 (12.2)	2 (10.5)
	Sufficient	1624 (96.4)	349 (89.0)	101 (87.8)	848 (86.9)	98 (67.6)	58 (63.7)	999 (93.3)	203 (84.9)	152 (84.0)	153 (91.1)	26 (72.2)	16 (72.7)	169 (92.3)	43 (87.8)	17 (89.5)
Age (y)	Mean (SD)	28.3 (7.6)	27.4 (5.3)	24.5 (5.3)	25.6 (8.0)	24.3 (5.6)	23.4 (4.8)	26.2 (7.9)	25.5 (5.5)	25.5 (5.5)	27.2 (7.7)	26.9 (6.1)	25.7 (6.3)	28.2 (7.8)	28.2 (4.8)	25.4 (4.8)
Poverty Income Rate	Mean (SD)	2.9 (1.6)	3.1 (1.6)	1.0 (0.5)	1.8 (1.5)	1.7 (1.4)	0.7 (0.5)	1.7 (1.3)	1.5 (1.2)	0.9 (0.5)	1.9 (1.4)	1.9 (1.6)	0.9 (0.5)	2.7 (1.7)	3.1 (1.9)	1.0 (0.5)

* pregnant and nonpregnant women.

## Data Availability

The serum retinol and demographic information of the Bronx dataset, originally published in the J Perinat Med (PMID:30231012), are available by request. The National Health and Nutrition Examination Survey (NHANES) datasets were downloaded from the NHANES repository (https://www.cdc.gov/nchs/nhanes/index.htm, accessed on 30 March 2020) and merged file in accordance with the NHANES guidelines and recommendations.

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
