# Peer review of "Disproportionate Vitamin A Deficiency in Women of Specific Ethnicities Linked to Differences in Allele Frequencies of Vitamin A-Related Polymorphisms"

_nutrients, 2021, doi:10.3390/nu13061743_

Round 1

Reviewer 1 Report

This manuscript reports findings from a study of the prevalence of vitamin A deficiency (VAD) in pregnant women from different ethnic groups living in the United States.  In addition, the authors tested for associations with vitamin A-related SNP allelic frequencies in each ethnic group.  The authors conclude from their data that the prevalence of VAD varies between different ethnic groups and races and that this may be casually associated with genetic variants conferring risk of low blood retinol levels.

The manuscript is well written and interesting.  The authors’ conclusions follow directly from their data.  The work and findings are very important for understanding population risk for developing VAD in the United States population.  Most importantly, the finding of genetic associations with VAD will need to be considered in the planning of interventions aimed at eradicating VAD.

This is an excellent piece of work.  However, I have two editorial concerns with the manuscript that I believe need to be fixed.

First, I most strongly believe that the authors need to stress the need for assessing frequencies of genetic variants in vitamin A processing genes prior to the start of nutritional interventions to better enable adequate food-based interventions.  This point is currently mentioned in the text, but it is buried in the text and may only be read by the most avid of readers.  This needs to be much more strongly and prominently stated in the abstract and elsewhere throughout the text.

On page 8, line 275, the authors comment on the elevation in serum retinyl ester levels observed for Mexican Americans over the last 4 decades.  There is a growing literature reporting similar observations in Africa and this has been proposed to reflect vitamin A toxicity.  Could this be the case for Mexican Americans?  This point needs to be considered more fully and thoroughly in the text.

Author Response

We thank the Reviewer for His/Her positive feedback and constructive suggestions. We agree with the points raised by the Reviewer and revised our manuscript accordingly. The changes are indicated in blue throughout the text.
We attached a file contains responses to each comment. Please see the attachment.

Reviewer 2 Report

  1. Page 2, line 82-83: ‘‘In this study, we focused on female participants aged 17 to 42 years’’
  • Are these also pregnant women? The title of the article is ‘‘Disproportionate vitamin A deficiency in pregnant women of specific ethnicities in the United States and ethnic differences in allele frequencies of polymorphisms of vitamin A-related genes’’
  1. Page 2, line 85: ‘‘After excluding those who did not provide serum retinol levels, ….a total of 4,082 study participants were included in the analysis’’, however in table 1, it was written that there were VAD = 40 and VAS = 27 giving a total of 67 participants. Accordingly, table 2, it was written that ‘‘VAD proportions of 4662 women..’’
  • How many women were included in the study? The participants included in this study should be the same participants included in the results section. The numbers of the participants should be consistent throughout the study.
  1. Page 4, table 1: the abbreviations should be clarified.

  1. The ethnic groupings are confusing. Table 2 showed Vitamin a deficiency (VAD) prevalence varies between different ethnic groups and races. Ethnic groups and races were defined as Non-Hispanic White, Non-Hispanic black, Mexican and Other Hispanic. However, in figure 2, the population were defined as Asian, Non-Hispanic White, Non-Hispanic Black, Mexican and Afro Caribbean. One is expected to be consequent on the definition of the ethnic groups and races throughout the study.

  1. The study should examine the vitamin A levels between the population based on the examined alleles. Population with lower serum-risk allele frequencies (rs10882272 C and rs738409) are then expected to have lower vitamin A levels compared to those with normal allele.

Author Response

(The authors gave the same response as above.)

Reviewer 3 Report

This paper studied the role of ethninicyty of retinol deficit using the measurement of alleles rs10882272 associated with deficit of RPB4  and Retinol

They studied with what ethnicity was associated with alleles rs10882272 more frequent in pregnant women of African or Afro-American origin, but less frequent in European and Asian ethnic groups
In the Bronx, black Americans of Mexican descent had a low rate of these alleles and less deficit in Retinol and Vitamin A.

This study shows that ethnicity influences retinol and vitamin A levels.

Author Response

We thank the Reviewer for His/Her positive feedback.

Round 2

Reviewer 2 Report

Page 14 (article):

Our re-analysis of 285 the Bronx study showed that more than 40% of pregnant women have serum retinol <0.70 μmol/L, strongly suggesting 286 that urgent actions with assessing frequencies of genetic variants in vitamin A processing genes need to be taken to reduce the VAD, especially in unusually susceptible ethnic groups, to reduce the risk of adverse health conditions of 288 the mother [51] and diseases of offspring later in life [47,52].

  • There is no evidence showing that assessing frequencies of lower serum-risk allele frequencies (rs10882272 C and rs738409) may reduce the VAD. Since this study didn't examine such association, referred association between these genetic variants in vitamin A processing genes and vitamin A levels should be included.

Author Response

We thank again this Reviewer for his/her constructive criticisms on our manuscript.

Page 14 (article):

Our re-analysis of the Bronx study showed that more than 40% of pregnant women have serum retinol <0.70 μmol/L, strongly suggesting that urgent actions with assessing frequencies of genetic variants in vitamin A processing genes need to be taken to reduce the VAD, especially in unusually susceptible ethnic groups, to reduce the risk of adverse health conditions of 288 the mother [51] and diseases of offspring later in life [47,52].

  • There is no evidence showing that assessing frequencies of lower serum-risk allele frequencies (rs10882272 C and rs738409) may reduce the VAD. Since this study didn't examine such association, referred association between these genetic variants in vitamin A processing genes and vitamin A levels should be included.

We fully understand the Reviewer’s concern which entirely stemmed from lack of clarity on our end. Indeed, in this study, we did not test directly the associations between the genetic variants and VAD status on our cohort. We have revised the sentence as follows: “Our re-analysis of the Bronx study showed that more than 40% of pregnant women have serum retinol <0.70 μmol/L, strongly suggesting that urgent actions need to be taken to reduce the VAD, especially in unusually susceptible ethnic groups, to reduce the risk of adverse health conditions of the mother [51] and diseases of offspring later in life [47,52]. Assessing genetic variants information prior to performing an effective nutrient supplementation could be a critical strategy to plan more effective food-based interventions to reduce VAD”.

Furthermore, we revised the Discussion section to provide clarification about the key studies we are referring to.

Page 12, line 268-270: “The authors were not able to test the association of these variants with VAD status as the rate of VAD in the individuals studied was low; however, they reported that the estimated relative difference in mean retinol levels per copy of the rs10882272 C allele from the overall meta-analysis was decreased of 3.0% [21].

Page 13, line 289-291: “A significant high correlation between serum concentrations of retinol and those of RBP4, even in the context of acute phase response and protein malnutrition, has been reported [41]. Therefore, the PNPLA3 I148M homozygotes may have lower serum retinol levels compared to 148I allele carriers”.
